# Block-Normalized Gradient Method: An Empirical Study for Training Deep Neural Network

## Abstract

In this paper, we propose a generic and simple strategy for utilizing stochastic gradient information in optimization. The technique essentially contains two consecutive steps in each iteration: 1) computing and normalizing each block (layer) of the mini-batch stochastic gradient; 2) selecting appropriate step size to update the decision variable (parameter) towards the negative of the block-normalized gradient. We conduct extensive empirical studies on various non-convex neural network optimization problems, including multi layer perceptron, convolution neural networks and recurrent neural networks. The results indicate the block-normalized gradient can help accelerate the training of neural networks. In particular, we observe that the normalized gradient methods having constant step size with occasionally decay, such as SGD with momentum, have better performance in the deep convolution neural networks, while those with adaptive step sizes, such as Adam, perform better in recurrent neural networks. Besides, we also observe this line of methods can lead to solutions with better generalization properties, which is confirmed by the performance improvement over strong baselines.

## 1 Introduction

Continuous optimization is a core technique for training non-convex sophisticated machine learning models such as deep neural networks (Bengio, 2009). Compared to convex optimization where a global optimal solution is expected, non-convex optimization usually aims to find a stationary point or a local optimal solution of an objective function by iterative algorithms. Among a large volume of optimization algorithms, first-order methods, which only iterate with the gradient information of objective functions, are widely used due to its relatively low requirement on memory space and computation time, compared to higher order algorithms. In many machine learning scenarios with large amount of data, the full gradient is still expensive to obtain, and hence the unbiased stochastic version will be adopted, as it is even more computationally efficient.

In this paper, we are particularly interested in solving the deep neural network training problems with stochastic first order methods. Compared to other non-convex problems, deep neural network training additionally has the following challenge: gradient may be vanishing and/or exploding. More specifically, due to the chain rule (a.k.a. backpropagation), the original gradient in the low layers will become very small or very large because of the multiplicative effect of the gradients from the upper layers, which is usually all smaller or larger than 1. As the number of layers in the neural network increases, the phenomenon of vanishing or exploding gradients becomes more severe such that the iterative solution will converge slowly or diverge quickly.

We aim to alleviate this problem by block-wise stochastic *gradient normalization*, which is constructed via dividing the stochastic gradient by its norm. Here, each block essentially contains the variables of one layer in a neural network, so it can also be interpreted as layer-wise gradient normalization. Compared to the regular gradient, normalized gradient only provides an updating direction but does not incorporate the local steepness of the objective through its magnitude, which helps to control the change of the solution through a well-designed step length. Intuitively, as it constrains the magnitude of the gradient to be 1, it should to some extent prevent the gradient vanishing or exploding phenomenon. In fact, as showed in (Hazan et al., 2015; Levy, 2016), normalized gra-

dient descent (NGD) methods are more numerically stable and have better theoretical convergence properties than the regular gradient descent method in non-convex optimization.

Once the updating direction is determined, step size (learning rate) is the next important component in the design of first-order methods. While for convex problems there are some well studied strategies to find a stepsize ensuring convergence, for non-convex optimization, the choice of step size is more difficult and critical as it may either enlarge or reduce the impact of the aforementioned vanishing or exploding gradients.

Among different choices of step sizes, the constant or adaptive *feature-dependent* step sizes are widely adopted. On one hand, stochastic gradient descent (SGD) + momentum + constant step size has become the standard choice for training feed-forward networks such as Convolution Neural Networks (CNN). Ad-hoc strategies like decreasing the step size when the validation curve plateaus are well adopted to further improve the generalization quality. On the other hand, different from the standard step-size rule which multiplies a same number to each coordinate of gradient, the adaptive feature-dependent step-size rule multiplies different numbers to coordinates of gradient so that different parameters in the learning model can be updated in different paces. For example, the adaptive step size invented by (Duchi et al., 2011) is constructed by aggregating each coordinates of historical gradients. As discussed by (Duchi et al., 2011), this method can dynamically incorporate the frequency of features in the step size so that frequently occurring coordinates will have a small step sizes while infrequent features have long ones. The similar adaptive step size is proposed in (Kingma & Ba, 2014) but the historical gradients are integrated into feature-dependent step size by a different weighting scheme.

In this paper, we propose a generic framework using the mini-batch stochastic normalized gradient as the updating direction (like (Hazan et al., 2015; Levy, 2016)) and the step size is either constant or adaptive to each coordinate as in (Duchi et al., 2011; Kingma & Ba, 2014). Our framework starts with computing regular mini-batch stochastic gradient, which is immediately normalized layer-wisely. The normalized version is then plugged in the constant stepsize with occasional decay, such as SGD+momentum, or the adaptive step size methods, such as Adam (Kingma & Ba, 2014) and AdaGrad (Duchi et al., 2011). The numerical results shows that normalized gradient always helps to improve the performance of the original methods especially when the network structure is deep. It seems to be the first thorough empirical study on various types of neural networks with this normalized gradient idea. Besides, although we focus our empirical studies on deep learning where the objective is highly non-convex, we also provide a convergence proof under this framework when the problem is convex and the stepsize is adaptive in the appendix. This convergence under the non-convex case will be a very interesting and important future work.

The rest of the paper is organized as follows. In Section 2, we briefly go through the previous work that are related to ours. In Section 3, we formalize the problem to solve and propose the generic algorithm framework. In Section 4, we conduct comprehensive experimental studies to compare the performance of different algorithms on various neural network structures. We conclude the paper in Section 5. Finally, in the appendix, we provide a concrete example of this type of algorithm and show its convergence property under the convex setting.

## 2 RELATED WORK

A pioneering work on normalized gradient descent (NGD) method was by Nesterov (Nesterov, 1984) where it was shown that NGD can find a $\epsilon$-optimal solution within $O(\frac{1}{\epsilon^2})$ iterations when the objective function is differentiable and quasi-convex. Kiwiel (Kiwiel, 2001) and Hazan et al (Hazan et al., 2015) extended NGD for upper semi-continuous (but not necessarily differentiable) quasi-convex objective functions and local-quasi-convex objective functions, respectively, and achieved the same iteration complexity. Moreover, Hazan et al (Hazan et al., 2015) showed that NGD's iteration complexity can be reduced to $O(\frac{1}{\epsilon})$ if the objective function is local-quasi-convex and locally-smooth. A stochastic NGD algorithm is also proposed by Hazan et al (Hazan et al., 2015) which, if a mini-batch is used to construct the stochastic normalized gradient in each iteration, finds $\epsilon$-optimal solution with a high probability for locally-quasi-convex functions within $O(\frac{1}{\epsilon^2})$ iterations. Levy (Levy, 2016) proposed a Saddle-Normalized Gradient Descent (Saddle-NGD) method, which adds a zero-mean Gaussian random noise to the stochastic normalized gradient periodically. When applied to strict-saddle functions with some additional assumption, it is shown (Levy, 2016)

that Saddle-NGD can evade the saddle points and find a local minimum point approximately with a high probability.

Analogous yet orthogonal to the gradient normalization ideas have been proposed for the deep neural network training. For example, batch normalization (Ioffe & Szegedy, 2015) is used to address the internal covariate shift phenomenon in the during deep learning training. It benefits from making normalization a part of the model architecture and performing the normalization for each training mini-batch. Weight normalization (Salimans & Kingma, 2016), on the other hand, aims at a reparameterization of the weight vectors that decouples the length of those weight vectors from their direction. Recently (Neyshabur et al., 2015) proposes to use path normalization, an approximate path-regularized steepest descent with respect to a path-wise regularizer related to max-norm regularization to achieve better convergence than vanilla SGD and AdaGrad. Perhaps the most related idea to ours is Gradient clipping. It is proposed in (Pascanu et al., 2013) to avoid the gradient explosion, by pulling the magnitude of a large gradient to a certain level. However, this method does not do anything when the magnitude of the gradient is small.

Adaptive step size has been studied for years in the optimization community. The most celebrated method is the line search scheme. However, while the exact line search is usually computational infeasible, the inexact line search also involves a lot of full gradient evaluation. Hence, they are not suitable for the deep learning setting. Recently, algorithms with adaptive step sizes start to be applied to the non-convex neural network training, such as AdaGrad (Duchi et al., 2012), Adam (Kingma & Ba, 2014) and RMSProp (Hinton et al.). However they directly use the unnormalized gradient, which is different from our framework. Singh et al. (2015) recently proposes to apply layer-wise specific step sizes, which differs from ours in that it essentially adds a term to the gradient rather than normalizing it. Recently Wilson et al. (2017) finds the methods with adaptive step size might converge to a solution with worse generalization. However, this is orthogonal to our focus in this paper.

## 3  ALGORITHM FRAMEWORK

In this section, we present our algorithm framework to solve the following general problem:

$$\min_{x \in \mathbb{R}^d} f(x) = \mathbb{E}(F(x, \xi)), \tag{1}$$

where $x = (x^1, x^2, \ldots, x^B) \in \mathbb{R}^d$ with $x^i \in \mathbb{R}^{d_i}$ and $\sum_{i=1}^{B} d_i = d$, $\xi$ is a random variable following some distribution $\mathbb{P}$, $F(\cdot, \xi)$ is a loss function for each $\xi$ and the expectation $\mathbb{E}$ is taken over $\xi$. In the case where (1) models an empirical risk minimization problem, the distribution $\mathbb{P}$ can be the empirical distribution over training samples such that the objective function in (1) becomes a finite-sum function. Now our goal is to minimize the objective function $f$ over $x$, where $x$ can be the parameters of a machine learning model when (1) corresponds to a training problem. Here, the parameters are partitioned into $B$ blocks. The problem of training a neural network is an important instance of (1), where each block of parameters $x^i$ can be viewed as the parameters associated to the $i$th layer in the network.

We propose the generic optimization framework in Algorithm 1. In iteration $t$, it firstly computes the partial (sub)gradient $F_i'(x_t, \xi_t)$ of $F$ with respect to $x^i$ for $i = 1, 2, \ldots$, at $x = x_t$ with a mini-batch data $\xi_t$, and then normalizes it to get a partial direction $g_t^i = \frac{F_i'(x_t, \xi_t)}{\|F_i'(x_t, \xi_t)\|_2}$. We define $g_t = (g_t^1, g_t^2, \ldots, g_t^B)$. The next is to find $d$ adaptive step sizes $\tau_t \in \mathbb{R}^d$ with each coordinate of $\tau_t$ corresponding to a coordinate of $x$. We also partition $\tau_t$ in the same way as $x$ so that $\tau_t = (\tau_t^1, \tau_t^2, \ldots, \tau_t^B) \in \mathbb{R}^B$ with $\tau_t^i \in \mathbb{R}^{d_i}$. We use $\tau_t$ as step sizes to update $x_t$ to $x_{t+1}$ as $x_{t+1} = x_t - \tau_t \circ g_t$, where $\circ$ represents coordinate-wise (Hadamard) product. In fact, our framework can be customized to most of existing first order methods with fixed or adaptive step sizes, such as SGD, AdaGrad(Duchi et al., 2011), RMSProp (Hinton et al.) and Adam(Kingma & Ba, 2014), by adopting their step size rules respectively.

## 4  NUMERICAL EXPERIMENTS

**Basic Experiment Setup**   In this section, we conduct comprehensive numerical experiments on different types of neural networks. The algorithms we are testing are SGD with Momentum

---

**Algorithm 1** Generic Block-Normalized Gradient (BNG) Descent

1: Choose $x_1 \in \mathbb{R}^d$.
2: **for** $t = 1, 2, ..., $ **do**
3:     Sample a mini-batch of data $\xi_t$ and compute the partial stochastic gradient $g_t^i = \frac{F_i'(x_t, \xi_t)}{\|F_i'(x_t, \xi_t)\|_2}$
4:     Let $g_t = (g_t^1, g_t^2, \ldots, g_t^B)$ and choose step sizes $\tau_t \in \mathbb{R}^d$.
5:     $x_{t+1} = x_t - \tau_t \circ g_t$
6: **end for**

---

(SGDM), AdaGrad (Duchi et al., 2013), Adam (Kingma & Ba, 2014) and their block-normalized gradient counterparts, which are denoted with suffix "NG". Specifically, we partition the parameters into block as $x = (x^1, x^2, \ldots, x^B)$ such that $x^i$ corresponds to the vector of parameters (including the weight matrix and the bias/intercept coefficients) used in the $i$th layer in the network.

Our experiments are on four diverse tasks, ranging from image classification to natural language processing. The neural network structures under investigation include multi layer perceptron, long-short term memory and convolution neural networks.

To exclude the potential effect that might be introduced by advanced techniques, in all the experiments, we only adopt the basic version of the neural networks, unless otherwise stated. The loss functions for classifications are cross entropy, while the one for language modeling is log perplexity. Since the computational time is proportional to the epochs, we only show the performance versus epochs. Those with running time are similar so we omit them for brevity. For all the algorithms, we use their default settings. More specifically, for Adam/AdamNG, the initial step size scale $\alpha = 0.001$, first order momentum $\beta_1 = 0.9$, second order momentum $\beta_2 = 0.999$, the parameter to avoid division of zero $\epsilon = 1e^{-8}$; for AdaGrad/AdaGradNG, the initial step size scale is 0.01.

### 4.1 MULTI LAYER PERCEPTRON FOR MNIST IMAGE CLASSIFICATION

The first network structure we are going to test upon is the Multi Layer Perceptron (MLP). We will adopt the handwritten digit recognition data set MNIST[1]Lecun et al. (1998), in which, each data is an image of hand written digits from $\{1, 2, 3, 4, 5, 6, 7, 8, 9, 0\}$. There are 60k training and 10k testing examples and the task is to tell the right number contained in the test image. Our approach is applying MLP to learn an end-to-end classifier, where the input is the raw $28 \times 28$ images and the output is the label probability. The predicted label is the one with the largest probability. In each middle layer of the MLP, the hidden unit number are 100, and the first and last layer respectively contain 784 and 10 units. The activation functions between layers are all sigmoid and the batch size is 100 for all the algorithms.

We choose different numbers of layer from $\{6, 12, 18\}$. The results are shown in Figure 1. Each column of the figures corresponds to the training and testing objective curves of the MLP with a given layer number. From left to right, the layer numbers are respectively 6, 12 and 18. We can see that, when the network is as shallow as containing 6 layers, the normalized stochastic gradient descent can outperform its unnormalized counterpart, while the Adam and AdaGrad are on par with or even slightly better than their unnormalized versions. As the networks become deeper, the acceleration brought by the gradient normalization turns more significant. For example, starting from the second column, AdamNG outperforms Adam in terms of both training and testing convergence. In fact, when the network depth is 18, the AdamNG can still converge to a small objective value while Adam gets stuck from the very beginning. We can observe the similar trend in the comparison between AdaGrad (resp. SGDM) and AdaGradNG (resp. SGDMNG). On the other hand, the algorithms with adaptive step sizes can usually generate a stable learning curve. For example, we can see from the last two column that SGDNG causes significant fluctuation in both training and testing curves. Finally, under any setting, AdamNG is always best algorithm in terms of convergence performance.

---

[1]http://yann.lecun.com/exdb/mnist/

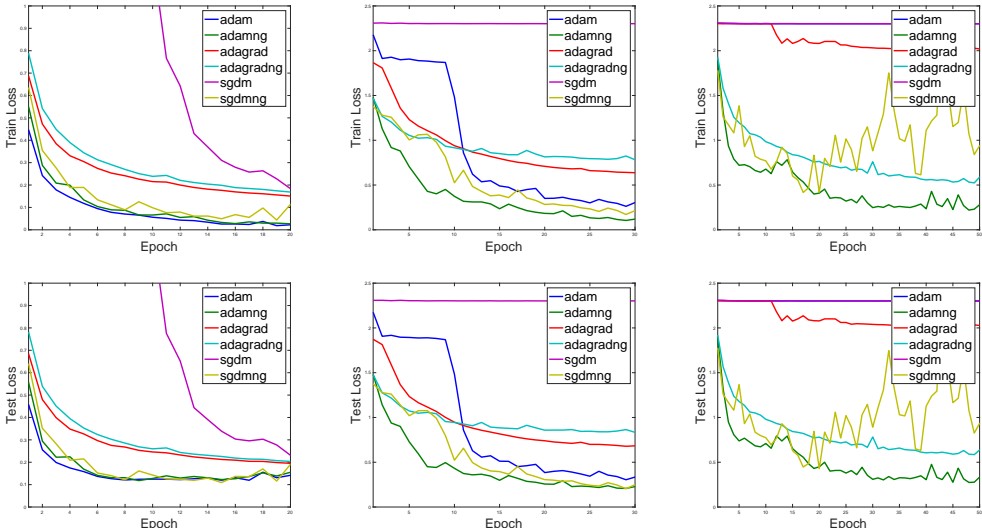

Figure 1: The training and testing objective curves on MNIST dataset with multi layer perceptron. From left to right, the layer numbers are 6, 12 and 18 respectively. The first row is the training curve and the second is testing.

## 4.2 RESIDUAL NETWORK ON CIFAR10 AND CIFAR100

**Datasets** In this section, we benchmark the methods on CIFAR (both CIFAR10 and CIFAR100) datasets with the residual networks He et al. (2016a), which consist mainly of convolution layers and each layer comes with batch normalization (Ioffe & Szegedy, 2015). CIFAR10 consists of 50,000 training images and 10,000 test images from 10 classes, while CIFAR100 from 100 classes. Each input image consists of $32 \times 32$ pixels. The dataset is preprocessed as described in He et al. (2016a) by subtracting the means and dividing the variance for each channel. We follow the same data augmentation in He et al. (2016a) that 4 pixels are padded on each side, and a $32 \times 32$ crop is randomly sampled from the padded image or its horizontal flip.

**Algorithms** We adopt two types of optimization frameworks, namely SGD and Adam[2], which respectively represent the constant step size and adaptive step size methods. We compare the performances of their original version and the layer-normalized gradient counterpart. We also investigate how the performance changes if the normalization is relaxed to not be strictly 1. In particular, we find that if the normalized gradient is scaled by its variable norm with a ratio, which we call NG$_{\text{adap}}$ and defined as follows,

$$\text{NG}_{\text{adap}} := \text{NG} \times \text{Norm of variable} \times \alpha = \text{Grad} \times \frac{\text{Norm of variable}}{\text{Norm of grad}} \times \alpha,$$

we can get lower testing error. The subscript "adap" is short for "adaptive", as the resulting norm of the gradient is adaptive to its variable norm, while $\alpha$ is the constant ratio. Finally, we also compare with the gradient clipping trick that rescales the gradient norm to a certain value if it is larger than that threshold. Those methods are with suffix "CLIP".

**Parameters** In the following, whenever we need to tune the parameter, we search the space with a holdout validation set containing 5000 examples.

For SGD+Momentum method, we follow exactly the same experimental protocol as described in He et al. (2016a) and adopt the publicly available Torch implementation[3] for residual network. In

---

[2]We also tried AdaGrad, but it has significantly worse performance than SGD and Adam, so we do not report its result here.

[3]https://github.com/facebook/fb.resnet.torch

particular, SGD is used with momentum of 0.9, weight decay of 0.0001 and mini-batch size of 128. The initial learning rate is 0.1 and dropped by a factor of 0.1 at 80, 120 with a total training of 160 epochs. The weight initialization is the same as He et al. (2015).

For Adam, we search the initial learning rate in range $\{0.0005, 0.001, 0.005, 0.01\}$ with the base algorithm Adam. We then use the best learning rate, i.e., 0.001, for all the related methods Adam-CLIP, AdamNG, and AdamNG$_{adap}$. Other setups are the same as the SGD. In particular, we also adopt the manually learning rate decay here, since, otherwise, the performance will be much worse.

We adopt the residual network architectures with depths $L = \{20, 32, 44, 56, 110\}$ on both CIFAR-10 and CIFAR100. For the extra hyper-parameters, i.e., threshold of gradient clipping and scale ratio of NG$_{adap}$, i.e., $\alpha$, we choose the best hyper-parameter from the 56-layer residual network. In particular, for clipping, the searched values are $\{0.05, 0.1, 0.5, 1, 5\}$, and the best value is 0.1. For the ratio $\alpha$, the searched values are $\{0.01, 0.02, 0.05\}$ and the best is 0.02.

**Results** For each network structure, we make 5 runs with random initialization. We report the training and testing curves on CIFAR10 and CIFAR100 datasets with deepest network Res-110 in Figure 2. We can see that the normalized gradient methods (with suffix "NG") converge the fastest in training, compared to the non-normalized counterparts. While the adaptive version NG$_{adap}$ is not as fast as NG in training, however, it can always converge to a solution with lower testing error, which can be seen more clearly in Table 1 and 2.

For further quantitative analysis, we report the means and variances of the final test errors on both datasets in Table 1 and 2, respectively. Both figures convey the following two messages. Firstly, on both datasets with ResNet, SGD+Momentum is uniformly better than Adam in that it always converges to a solution with lower test error. Such advantage can be immediately seen by comparing the two row blocks within each column of both tables. It is also consistent with the common wisdom (Wilson et al., 2017). Secondly, for both Adam and SGD+Momentum, the NG$_{adap}$ version has the best generalization performance (which is mark bold in each column), while the gradient clipping is inferior to all the remaining variants. While the normalized SGD+Momentum is better than the vanilla version, Adam slightly outperforms its normalized counterpart. Those observations are consistent across networks with variant depths.

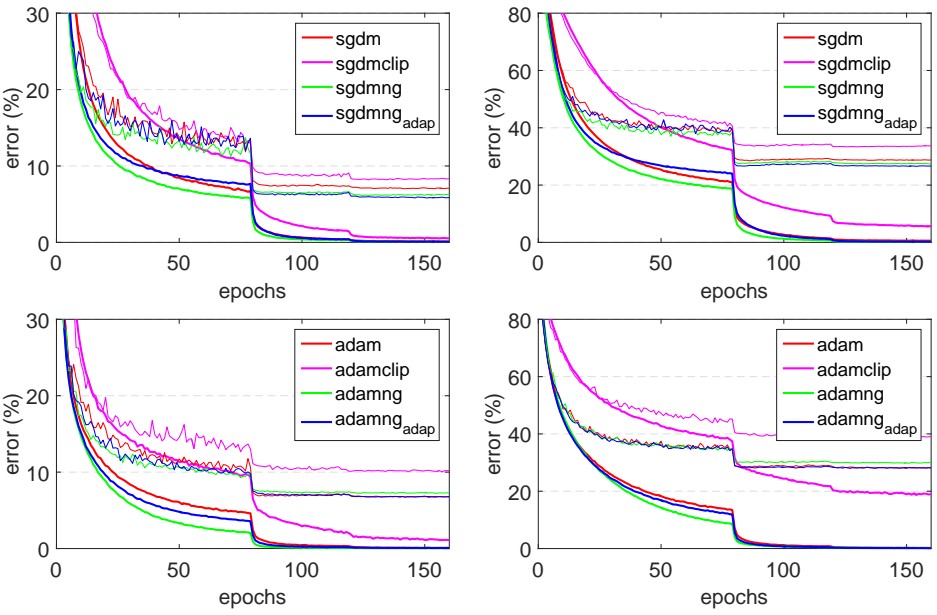

Figure 2: The training and testing curves on CIFAR10 and CIFAR100 datasets with Resnet-110. Left: CIFAR10; Right: CIFAR100; Upper: SGD+Momentum; Lower: Adam. The thick curves are the training while the thin are testing.

| Algorithm | ResNet-20 | ResNet-32 | ResNet-44 | ResNet-56 | ResNet-110 |
|---|---|---|---|---|---|
| | | | Adam | | |
| Adam | $9.14 \pm 0.07$ | $8.33 \pm 0.17$ | $7.794 \pm 0.22$ | $7.33 \pm 0.19$ | $6.75 \pm 0.30$ |
| AdamCLIP | $10.18 \pm 0.16$ | $9.18 \pm 0.06$ | $8.89 \pm 0.14$ | $9.24 \pm 0.19$ | $9.96 \pm 0.29$ |
| AdamNG | $9.42 \pm 0.20$ | $8.50 \pm 0.17$ | $8.06 \pm 0.20$ | $7.69 \pm 0.19$ | $7.29 \pm 0.08$ |
| AdamNG$_{adap}$ | $\mathbf{8.52 \pm 0.16}$ | $\mathbf{7.62 \pm 0.25}$ | $\mathbf{7.28 \pm 0.18}$ | $\mathbf{7.04 \pm 0.27}$ | $\mathbf{6.71 \pm 0.17}$ |
| | | | SGD+Momentum | | |
| SGDM* | 8.75 | 7.51 | 7.17 | 6.97 | $6.61 \pm 0.16$ |
| SGDM | $7.93 \pm 0.15$ | $7.15 \pm 0.20$ | $7.09 \pm 0.21$ | $7.34 \pm 0.52$ | $7.07 \pm 0.65$ |
| SGDMCLIP | $9.03 \pm 0.15$ | $8.44 \pm 0.14$ | $8.55 \pm 0.20$ | $8.30 \pm 0.08$ | $8.35 \pm 0.25$ |
| SGDMNG | $7.82 \pm 0.26$ | $7.09 \pm 0.13$ | $6.60 \pm 0.21$ | $6.59 \pm 0.23$ | $6.28 \pm 0.22$ |
| SGDMNG$_{adap}$ | $\mathbf{7.71 \pm 0.18}$ | $\mathbf{6.90 \pm 0.11}$ | $\mathbf{6.43 \pm 0.03}$ | $\mathbf{6.19 \pm 0.11}$ | $\mathbf{5.87 \pm 0.10}$ |

Table 1: Error rates of ResNets with different depths on CIFAR 10. SGDM* indicates the results reported in He et al. (2015) with the same experimental setups as ours, where only ResNet-110 has multiple runs.

| Algorithm | ResNet-20 | ResNet-32 | ResNet-44 | ResNet-56 | ResNet-110 |
|---|---|---|---|---|---|
| | | | Adam | | |
| Adam | $34.44 \pm 0.33$ | $32.94 \pm 0.16$ | $31.53 \pm 0.13$ | $30.80 \pm 0.30$ | $28.20 \pm 0.14$ |
| AdamCLIP | $38.10 \pm 0.48$ | $35.78 \pm 0.20$ | $35.41 \pm 0.19$ | $35.62 \pm 0.39$ | $39.10 \pm 0.35$ |
| AdamNG | $35.06 \pm 0.39$ | $33.78 \pm 0.07$ | $32.26 \pm 0.29$ | $31.86 \pm 0.21$ | $29.87 \pm 0.49$ |
| AdamNG$_{adap}$ | $\mathbf{32.98 \pm 0.52}$ | $\mathbf{31.74 \pm 0.07}$ | $\mathbf{30.75 \pm 0.60}$ | $\mathbf{29.92 \pm 0.26}$ | $\mathbf{28.09 \pm 0.46}$ |
| | | | SGD+Momentum | | |
| SGDM | $32.28 \pm 0.16$ | $30.62 \pm 0.36$ | $29.96 \pm 0.66$ | $29.07 \pm 0.41$ | $28.79 \pm 0.63$ |
| SGDMCLIP | $35.06 \pm 0.37$ | $34.49 \pm 0.49$ | $33.36 \pm 0.36$ | $34.00 \pm 0.96$ | $33.38 \pm 0.73$ |
| SGDMNG | $32.46 \pm 0.37$ | $31.16 \pm 0.37$ | $30.05 \pm 0.29$ | $29.42 \pm 0.51$ | $27.49 \pm 0.25$ |
| SGDMNG$_{adap}$ | $\mathbf{31.43 \pm 0.35}$ | $\mathbf{29.56 \pm 0.25}$ | $\mathbf{28.92 \pm 0.28}$ | $\mathbf{28.48 \pm 0.19}$ | $\mathbf{26.72 \pm 0.39}$ |

Table 2: Error rates of ResNets with different depths on CIFAR 100. Note that He et al. (2015) did not run experiment on CIFAR 100.

### 4.3 RESIDUAL NETWORK FOR IMAGENET CLASSIFICATION

In this section, we further test our methods on ImageNet 2012 classification challenges, which consists of more than 1.2M images from 1,000 classes. We use the given 1.28M labeled images for training and the validation set with 50k images for testing. We employ the validation set as the test set, and evaluate the classification performance based on top-1 and top-5 error. The pre-activation (He et al., 2016b) version of ResNet is adopted in our experiments to perform the classification task. Like the previous experiment, we again compare the performance on SGD+Momentum and Adam.

We run our experiments on one GPU and use single scale and single crop test for simplifying discussion. We keep all the experiments settings the same as the publicly available Torch implementation [4]. That is, we apply stochastic gradient descent with momentum of 0.9, weight decay of 0.0001, and set the initial learning rate to 0.1. The exception is that we use mini-batch size of 64 and 50 training epochs considering the GPU memory limitations and training time costs. Regarding learning rate annealing, we use 0.001 exponential decay.

As for Adam, we search the initial learning rate in range $\{0.0005, 0.001, 0.005, 0.01\}$. Other setups are the same as the SGD optimization framework. Due to the time-consuming nature of training the networks (which usually takes one week) in this experiment, we only test on a 34-layer ResNet and compare SGD and Adam with our default NG method on the testing error of the classification. From Table 3, we can see normalized gradient has a non-trivial improvement on the testing error over the baselines SGD and Adam. Besides, the SGD+Momentum again outperforms Adam, which

---

[4]We again use the public Torch implementation: `https://github.com/facebook/fb.resnet.torch`

is consistent with both the common wisdom (Wilson et al., 2017) and also the findings in previous section.

| method | Top-1 | Top-5 |
|--------|-------|-------|
| Adam | 35.6 | 14.09 |
| AdamNG | 30.17 | 10.51 |
| SGDM | 29.05 | 9.95 |
| SGDMNG | **28.43** | **9.57** |

Table 3: Top-1 and Top 5 error rates of ResNet on ImageNet classification with different algorithms.

## 4.4 LANGUAGE MODELING WITH RECURRENT NEURAL NETWORK

Now we move on to test the algorithms on Recurrent Neural Networks (RNN). In this section, we test the performance of the proposed algorithm on the word-level language modeling task with a popular type of RNN, i.e. single directional Long-Short Term Memory (LSTM) networks (Hochreiter & Schmidhuber, 1997). The data set under use is Penn Tree Bank (PTB) (Marcus et al., 1993) data, which, after preprocessed, contains 929k training words, 73k validation and 82k test words. The vocabulary size is about 10k. The LSTM has 2 layers, each containing 200 hidden units. The word embedding has 200 dimensions which is trained from scratch. The batch size is 100. We vary the length of the backprop through time (BPTT) within the range $\{40, 400, 1000\}$. To prevent overfitting, we add a dropout regularization with rate 0.5 under all the settings.

The results are shown in Figure 3. The conclusions drawn from those figures are again similar to those in the last two experiments. However, the slightly different yet cheering observations is that the AdamNG is uniformly better than all the other competitors with any training sequence length. The superiority in terms of convergence speedup exists in both training and testing.

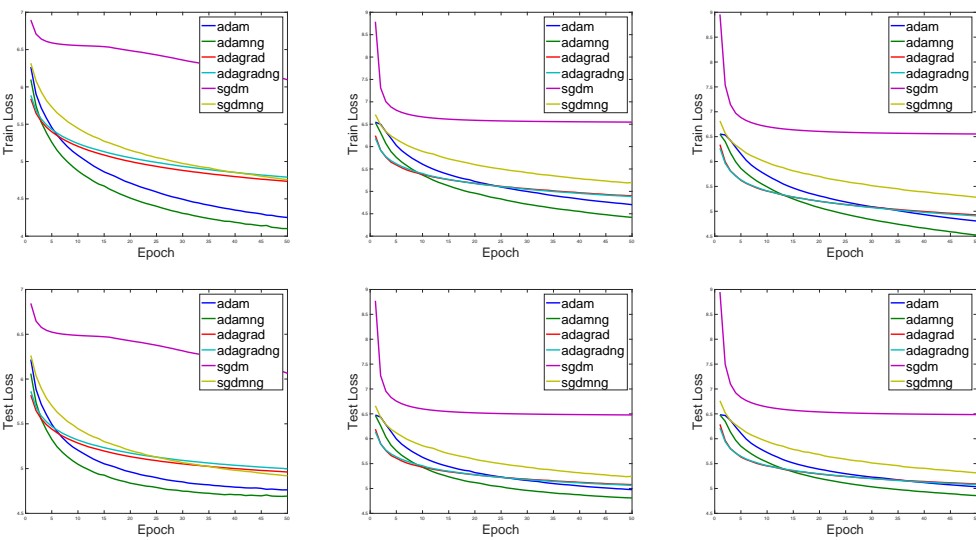

Figure 3: The training and testing objective curves on Penn Tree Bank dataset with LSTM recurrent neural networks. The first row is the training objective while the second is the testing. From left to right, the training sequence (BPTT) length are respectively 40, 400 and 1000. Dropout with 0.5 is imposed.

### 4.5 SENTIMENT ANALYSIS WITH CONVOLUTION NEURAL NETWORK

The task in this section is the sentiment analysis with convolution neural network. The dataset under use is Rotten Tomatoes[5] Pang & Lee (2005), a movie review dataset containing 10,662 documents, with half positive and half negative. We randomly select around 90% for training and 10% for validation. The model is a single layer convolution neural network that follows the setup of (Kim, 2014). The word embedding under use is randomly initialized and of 128-dimension.

For each algorithm, we run 150 epochs on the training data, and report the best validation accuracy in Table 4. The messages conveyed by the table is three-fold. Firstly, the algorithms using normalized gradient achieve much better validation accuracy than their unnormalized versions. Secondly, those with adaptive stepsize always obtain better accuracy than those without. This is easily seen by the comparison between Adam and SGDM. The last point is the direct conclusion from the previous two that the algorithm using normalized gradient with adaptive step sizes, namely AdamNG, outperforms all the remaining competitors.

| Algorithm | AdamNG | Adam | AdaGradNG | AdaGrad | SGDMNG | SGDM |
|---|---|---|---|---|---|---|
| Validation Accuracy | **77.11%** | 74.02% | 71.95% | 69.89% | 71.95% | 64.35% |

Table 4: The Best validation accuracy achieved by the different algorithms.

## 5 CONCLUSION

In this paper, we propose a generic algorithm framework for first order optimization. It is particularly effective for addressing the vanishing and exploding gradient challenge in training with non-convex loss functions, such as in the context of convolutional and recurrent neural networks. Our method is based on normalizing the gradient to establish the descending direction regardless of its magnitude, and then separately estimating the ideal step size adaptively or constantly. This method is quite general and may be applied to different types of networks and various architectures. Although the primary application of the algorithm is deep neural network training, we provide a convergence for the new method under the convex setting in the appendix.

Empirically, the proposed method exhibits very promising performance in training different types of networks (convolutional, recurrent) across multiple well-known data sets (image classification, natural language processing, sentiment analysis, etc.). In general, the positive performance differential compared to the baselines is most striking for very deep networks, as shown in our comprehensive experimental study.

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

APPENDIX

As a concrete example for Algorithm 1, we present a simple modification of AdaGrad using block-wise normalized stochastic gradient in Algorithm 2, where $g_{1:t}$ is a matrix created by stacking $g_1$, $g_2$,... and $g_t$ in columns and $g_{1:t,j} \in \mathbb{R}^t$ represents the $j$th row of $g_{1:t}$ for $j = 1, 2, \ldots, d$. Assuming the function $F$ is convex over $x$ for any $\xi$ and following the analysis in (Duchi et al., 2011), it is straightforward to show a $O(\frac{1}{\sqrt{T}})$ convergence rate of this modification. In the following, we denote $\|x\|_W := \sqrt{x^\top W x}$ as the the Mahalanobis norm associated to a $d \times d$ positive definite matrix $W$. The convergence property of Block-Normalized AdaGrad is presented in the following theorem.

---

**Algorithm 2** AdaGrad with Block-Normalized Gradient (AdaGradBNG)

---

1: Choose $x_1 \in \mathbb{R}^d$, $\delta > 0$ and $\eta > 0$.
2: **for** $t = 1, 2, ...,$ **do**
3:  Sample a mini-batch data $\xi_t$ and compute the stochastic partial gradient $g_t^i = \frac{F_i'(x_t, \xi_t)}{\|F_i'(x_t, \xi_t)\|_2}$
4:  Let $g_t = (g_t^1, g_t^2, \ldots, g_t^B)$, $g_{1:t} = [g_1, g_2, \ldots, g_t]$ and $s_t = (\|g_{1:t,1}\|_2, \|g_{1:t,2}\|_2, \ldots, \|g_{1:t,d}\|_2)$
5:  Partition $s_t = (s_t^1, s_t^2, \ldots, s_t^B)$ in the same way as $g_t$.
6:  Let $\tau_t = (\tau_t^1, \tau_t^2, \ldots, \tau_t^B)$ with[6] $\tau_t^i = \eta \|F_i'(x_t, \xi_t)\|_2 (\delta \mathbf{1}_{d_i} + s_t^i)^{-1}$.
7:  $x_{t+1} = x_t - \tau_t \circ g_t$
8: **end for**

---

**Theorem 1** *Suppose $F$ is convex over $x$, $\|F_i'(x_t, \xi_t)\|_2 \leq M_i$ and $\|x_t - x^*\|_\infty \leq D_\infty$ for all $t$ for some constants $M_i$ and $D_\infty > 0$ in Algorithm 2. Let $H_t = \delta I_d + diag(s_t)$ and $H_t^i = \delta I_{d_i} + diag(s_t^i)$ for $t = 1, 2, \ldots$ and $\bar{x}_T := \frac{1}{T} \sum_{t=1}^T x_t$. Algorithm 2 guarantees*

$$
\mathbb{E}[f(\bar{x}_T) - f(x^*)] \leq \frac{\|x_1 - x^*\|_{H_1}^2}{2\eta T} + \frac{D_\infty^2 \sqrt{Bd}}{2\eta \sqrt{T}} + \sum_{i=1}^B \frac{\eta \mathbb{E}\left[M_i^2 \sum_{j=d_1+d_2+\cdots+d_{i-1}+1}^{d_1+d_2+\cdots+d_i} \|g_{1:T,j}\|_2\right]}{T}
$$

$$
\leq \frac{\|x_1 - x^*\|_{H_1}^2}{2\eta T} + \frac{D_\infty^2 \sqrt{Bd}}{2\eta \sqrt{T}} + \sum_{i=1}^B \frac{\eta M_i^2 \sqrt{d_i}}{\sqrt{T}}.
$$

*Proof:* It is easy to see that $H_t$ is a positive definite and diagonal matrix. According to the updating scheme of $x_{t+1}$ and the definitions of $H_t$, $\tau_t$ and $g_t$, we have

$$
\begin{aligned}
\|x_{t+1} - x^*\|_{H_t}^2 &= (x_t - \tau_t \circ g_t - x^*)^\top H_t (x_t - \tau_t \circ g_t - x^*) \\
&= \|x_t - x^*\|_{H_t}^2 - 2(x_t - x^*)^\top H_t(\tau_t \circ g_t) + \|\tau_t \circ g_t\|_{H_t}^2 \\
&\leq \|x_t - x^*\|_{H_t}^2 - 2\eta(x_t - x^*)^\top F'(x_t, \xi_t) + \|\tau_t \circ g_t\|_{H_t}^2.
\end{aligned}
$$

The inequality above and the convexity of $F(x, \xi)$ in $x$ imply

$$
\begin{aligned}
F(x_t, \xi_t) - F(x^*, \xi_t) &\leq \frac{\|x_t - x^*\|_{H_t}^2}{2\eta} - \frac{\|x_{t+1} - x^*\|_{H_t}^2}{2\eta} + \frac{\|\tau_t \circ g_t\|_{H_t}^2}{2\eta} \\
&= \frac{\|x_t - x^*\|_{H_t}^2}{2\eta} - \frac{\|x_{t+1} - x^*\|_{H_t}^2}{2\eta} + \sum_{i=1}^B \frac{\eta \|F_i'(x_t, \xi_t)\|_2^2 \|g_t^i\|_{(H_t^i)^{-1}}^2}{2} \quad (2)
\end{aligned}
$$

Taking expectation over $\xi_t$ for $t = 1, 2, \ldots$ and averaging the above inequality give

$$
\begin{aligned}
&\mathbb{E}[f(\bar{x}_T) - f(x^*)] \\
&\leq \frac{1}{T} \sum_{t=1}^T \left[ \frac{\mathbb{E}\|x_t - x^*\|_{H_t}^2}{2\eta} - \frac{\mathbb{E}\|x_{t+1} - x^*\|_{H_t}^2}{2\eta} \right] + \sum_{t=1}^T \sum_{i=1}^B \frac{\eta \mathbb{E}\left[\|F_i'(x_t, \xi_t)\|_2^2 \|g_t^i\|_{(H_t^i)^{-1}}^2\right]}{2T} \\
&\leq \frac{1}{T} \sum_{t=1}^T \left[ \frac{\mathbb{E}\|x_t - x^*\|_{H_t}^2}{2\eta} - \frac{\mathbb{E}\|x_{t+1} - x^*\|_{H_t}^2}{2\eta} \right] + \sum_{t=1}^T \sum_{i=1}^B \frac{\eta M_i^2 \mathbb{E}\left[\|g_t^i\|_{(H_t^i)^{-1}}^2\right]}{2T}, \quad (3)
\end{aligned}
$$

where we use the fact that $\|F_i'(x_t, \xi_t)\|_2^2 \leq M_i^2$ in the second inequality.

According to the equation (24) in the proof of Lemma 4 in (Duchi et al., 2011), we have

$$\sum_{t=1}^{T} \|g_t^i\|_{(H_t^i)^{-1}}^2 = \sum_{t=1}^{T} \sum_{j=d_1+d_2+\cdots+d_{i-1}+1}^{d_1+d_2+\cdots+d_i} \frac{g_{t,j}^2}{\delta + \|g_{1:t,j}\|_2} \leq \sum_{j=d_1+d_2+\cdots+d_{i-1}+1}^{d_1+d_2+\cdots+d_i} 2\|g_{1:T,j}\|_2. \quad (4)$$

Following the analysis in the proof of Theorem 5 in (Duchi et al., 2011), we show that

$$\|x_{t+1} - x^*\|_{H_{t+1}}^2 - \|x_{t+1} - x^*\|_{H_t}^2 = \langle x^* - x_{t+1}, \mathrm{diag}(s_{t+1} - s_t)(x^* - x_{t+1}) \rangle$$
$$\leq D_\infty^2 \|s_{t+1} - s_t\|_1 = D_\infty^2 \langle s_{t+1} - s_t, \mathbf{1} \rangle. \quad (5)$$

After applying (4) and (5) to (3) and reorganizing terms, we have

$$\mathbb{E}[f(\bar{x}_T) - f(x^*)]$$
$$\leq \frac{\|x_1 - x^*\|_{H_1}^2}{2\eta T} + \frac{D_\infty^2 \mathbb{E}\langle s_T, \mathbf{1} \rangle}{2\eta T} + \sum_{i=1}^{B} \frac{\eta \mathbb{E}\left[ M_i^2 \sum_{j=d_1+d_2+\cdots+d_{i-1}+1}^{d_1+d_2+\cdots+d_i} \|g_{1:T,j}\|_2 \right]}{T}$$
$$\leq \frac{\|x_1 - x^*\|_{H_1}^2}{2\eta T} + \frac{D_\infty^2 \sqrt{Bd}}{2\eta\sqrt{T}} + \sum_{i=1}^{B} \frac{\eta \mathbb{E}\left[ M_i^2 \sum_{j=d_1+d_2+\cdots+d_{i-1}+1}^{d_1+d_2+\cdots+d_i} \|g_{1:T,j}\|_2 \right]}{T},$$

where the second inequality is because $\langle s_T, \mathbf{1} \rangle = \sum_{j=1}^{d} \|g_{1:T,j}\|_2 \leq \sqrt{TBd}$ which holds due to Cauchy-Schwarz inequality and the fact that $\|g_t^i\|_2 = 1$. Then, we obtain the first inequality in the conclusion of the theorem. To obtain the second inequality, we only need to observe that

$$\sum_{j=d_1+d_2+\cdots+d_{i-1}+1}^{d_1+d_2+\cdots+d_i} \|g_{1:T,j}\|_2 \leq \sqrt{Td_i} \quad (6)$$

which holds because of Cauchy-Schwarz inequality and the fact that $\|g_t^i\|_2 = 1$. ∎

**Remark 1** *When $B = 1$, namely, the normalization is applied to the full gradient instead of different blocks of the gradient, the inequality in Theorem 1 becomes*

$$\mathbb{E}[f(\bar{x}_T) - f(x^*)] \leq \frac{\|x_1 - x^*\|_{H_1}^2}{2\eta T} + \frac{D_\infty^2 \sqrt{d}}{2\eta\sqrt{T}} + \frac{\eta M^2 \sqrt{d}}{\sqrt{T}},$$

*where $M$ is a constant such that $\|F'(x_t, \xi_t)\|_2 \leq M$. Note that the right hand side of this inequality can be larger than that of the inequality in Theorem 1 with $B > 1$. We use $B = 2$ as an example. Suppose $F_1'$ dominates in the norm of $F'$, i.e., $M_2 \ll M_1 \approx M$ and $d_1 \ll d_2 \approx d$, we can have $\sum_{i=1}^{B} M_i^2 \sqrt{d_i} = O(M^2 \sqrt{d_1} + M_2^2 \sqrt{d})$ which can be much smaller than the factor $M^2 \sqrt{d}$ in the inequality above, especially when $M$ and $d$ are both large. Hence, the optimal value for $B$ is not necessarily one.*

