# OpenReview forum: "BLOCK-NORMALIZED GRADIENT METHOD: AN EMPIRICAL STUDY FOR TRAINING DEEP NEURAL NETWORK"
_ICLR.cc/2018/Conference — Reject_

### Official Review · AnonReviewer2 · 2017-11-27
**Clearly written paper, but experiments are not compelling and theoretical result is suboptimal**

**Rating:** 4
**Confidence:** 5

**Review:**

This paper proposes a family of first-order stochastic optimization schemes based on (1)  normalizing (batches of) stochastic gradient descents and (2) choosing from a step size updating scheme. The authors argue that iterative first-order optimization algorithms can be interpreted as a choice of an update direction and a step size, so they suggest that one should always normalize the gradient when computing the direction and then choose a step size using the normalized gradient.

The presentation in the paper is clear, and the exposition is easy to follow. The authors also do a good job of presenting related work and putting their ideas in the proper context. The authors also test their proposed method on many datasets, which is appreciated.

However, I didn't find the main idea of the paper to be particularly compelling. The proposed technique is reasonable on its own, but the empirical results do not come with any measure of statistical significance. The authors also do not analyze the sensitivity of the different optimization algorithms to hyperparameter choice, opting to only use the default. Moreover, some algorithms were used as benchmarks on some datasets but not others. For a primarily empirical paper, every state-of-the-art algorithm should be used as a point of comparison on every dataset considered. These factors altogether render the experiments uninformative in comparing the proposed suite of algorithms to state-of-the-art methods. The theoretical result in the convex setting is also not data-dependent, despite the fact that it is the normalized gradient version of AdaGrad, which does come with a data-dependent convergence guarantee.

Given the suite of optimization algorithms in the literature and in use today, any new optimization framework should either demonstrate improved (or at least matching) guarantees in some common (e.g. convex) settings or definitively outperform state-of-the-art methods on problems that are of widespread interest. Unfortunately, this paper does neither.

Because of these points, I do not feel the quality, originality, and significance of the work to be high enough to merit acceptance.

Some specific comments:
p. 2: "adaptive feature-dependent step size has attracted lots of attention". When you apply feature-dependent step sizes, you are effectively changing the direction of the gradient, so your meta learning formulation, as posed, doesn't make as much sense.
p. 2: "we hope the resulting methods can benefit from both techniques". What reason do you have to hope for this? Why should they be complimentary? Existing optimization techniques are based on careful design and coupling of gradients or surrogate gradients, with specific learning rate schedules. Arbitrarily mixing the two doesn't seem to be theoretically well-motivated.
p. 2: "numerical results shows that normalized gradient always helps to improve the performance of the original methods when the network structure is deep". It would be great to provide some intuition for this.
p. 2: "we also provide a convergence proof under this framework when the problem is convex and the stepsize is adaptive". The result that you prove guarantees a \theta(\sqrt{T}) convergence rate. On the other hand, the AdaGrad algorithm guarantees a data-dependent bound that is O(\sqrt{T}) but can also be much smaller. This suggests that there is no theoretical motivation to use NGD with an adaptive step size over AdaGrad.
p. 2-3: "NGD can find a \eps-optimal solution....when the objective function is quasi-convex. ....extended NGD for upper semi-continuous quasi-convex objective functions...". This seems like a typo. How are results that go from quasi-convex to upper semi-continuous quasi-convex an extension?
p. 3: There should be a reference for RMSProp.
p. 3: "where each block of parameters x^i can be viewed as parameters associated to the ith layer in the network". Why is layer parametrization (and later on normalization) a good way idea? There should be either a reference or an explanation.
p. 4: "x=(x_1, x_2, \ldots, x_B)". Should these subscripts be superscripts?
p. 4: "For all the algorithms, we use their default settings." This seems insufficient for an empirical paper, since most problems often involve some amount of hyperparameter tuning. How sensitive is each method to the choice of hyperparameters? What about the impact of initialization?
p. 4-8: None of the experimental results have error bars or any measure of statistical significance.
p. 5: "NG... is a variant of the NG_{UNIT} method". This method is never motivated.
p. 5-6: Why are SGD and Adam used for MNIST but not on CIFAR?
p. 5: "we chose the best heyper-paerameter from the 56 layer residual network." Apart from the typos, are these parameters chosen from the training set or the test set?
p. 6: Why isn't Adam tested on ImageNet?


POST AUTHOR RESPONSE: After reading the author response and taking into account the fact that the authors have spent the time to add more experiments and clarify their theoretical result, I have decided to upgrade my score from a 3 to a 4. However, I still do not feel that the paper is up to the standards of the conference.

---

> ### Author Response · Authors · 2018-01-05
> **Response to AnonReviewer2 (part 1)**
>
> We first thank the reviewer for the valuable feedback. The responses to your questions are as follows:
>
> Q: “The proposed technique is reasonable on its own, but the empirical results do not come with any measure of statistical significance.”
> A: Thank you for the suggestion! We have included the mean and variance in our experimental results in Section 4.2. Please see the revised version. We show that the normalized gradient method is indeed better than its unnormalized counterpart in many scenarios and it is not by chance.
>
> Q: “For a primarily empirical paper, EVERY state-of-the-art algorithm should be used as a point of comparison on EVERY dataset considered.”
> A: We believe this is a very harsh and unrealistic requirement and strikingly contradicts with AnonReviewer4’s suggestion. Nowadays, getting state-of-the-art performance on a dataset usually appeals to the combination of different efforts, including data preprocessing/augmentation, careful model designing and thorough parameter tuning, etc. However, it is not the main focus of this paper. Our paper aims to provide a simple alternative to train neural networks and it empirically works well on a number of tasks. In fact, in the CIFAR10/100 and ImageNet experiment, we largely adopted the parameter settings in [1], where the model championed the ImageNet 2015 challenge, except for the layer number, mini-batch size and GPU number (we don’t have that many GPUs). This should be considered a very strong baseline. Those parameters were well tuned by other researchers and we don’t see the necessity for re-tuning. Furthermore, we NEVER claim our method is a panacea and we believe none of the existing methods are either.
>
> [1] Kaiming He, Xiangyu Zhang, Shaoqing Ren, Jian Sun, Deep Residual Learning for Image Recognition. CVPR 2016.
>
> Q: “The theoretical result in the convex setting is also not data-dependent, despite the fact that it is the normalized gradient version of AdaGrad, which does come with a data-dependent convergence guarantee.”
> A: We are not sure what “data-dependent” means in the reviewer’s question. Since the reviewer said AdaGrad has a data-dependent convergence guarantee, we compare the convergence result of AdaGrad with our Theorem 1. We guess that the reviewer probably meant that the right-hand side of Adagrad’s convergence result (the inequalities in Theorem 5 by Duchi et al(2011)) has a summation of the norms of rows of historical gradients,, i.e., $\sum_{i=1}^d \|g_{1:T,i}\|_2$, which makes their bounds data-dependent. If our understanding is correct, we think our convergence guarantee is in fact data-dependent in the exactly the same way as AdaGrad. Please take a look at our inequality (3) which contains two components: the first component depends on $x_t$ and the second component depends on $\|g_t^i\|$. The first component is still upper bounded as in (5) but the second component can be bounded just by the first inequality in (4). Then our convergence guarantee will contain  $\sum_{i=1}^d \|g_{1:T,i}\|_2$ and will have the same data-dependency as AdaGrad. The reason this dependency did not appear in our Theorem 1 is because we further upper bounded the second component by 2\sqrt{Td_i} as we showed in the second inequality in (4). In fact, the authors of AdaGrad also did the same thing in their Corollary 6 where the data-dependent term $\sum_{i=1}^d \|g_{1:T,i}\|_2$ were also upper bounded by simpler terms. We thought the bound we reported in Theorem 1 was simpler. In the revision, we have included the data-dependent bound in Theorem 1.
>
> Q: “Any new optimization framework should either demonstrate improved (or at least matching) guarantees in some common (e.g. convex) settings or definitively outperform state-of-the-art methods on problems that are of widespread interest.”
> A: In terms of the performance of optimization under the convex setting, our result (Theorem 1) indeed matches AdaGrad in many ways. First, the optimality gaps ensured by both AgaGrad and our method convergence to zero in a rate of $1/\sqrt{T}$. Second, according to our answer to the last question, the convergence guarantee of both AgaGrad and our method are data-dependent and contain the term $\sum_{i=1}^d \|g_{1:T,i}\|_2$ in the same way. (We further upper bounded this term by a simpler quantity so this data-dependency might not be observed directly.) In addition, our method generalize AdaGrad by using block-wise adaptive subgradient.

---

> ### Author Response · Authors · 2018-01-05
> **Response to AnonReviewer2 (part 2)**
>
> Q: “When you apply feature-dependent step sizes, you are effectively changing the direction of the gradient, so your meta learning formulation, as posed, doesn't make as much sense.”
> A: We agree that we are indeed changing the direction of the real gradient. However, in this work we do demonstrate that this modification works well. We should also point out that a number of very successful and widely used approaches, such as batch normalization, layer normalization, weight normalization, gradient clipping, do dynamically change the data, or the weight, or the direction of the gradient. We believe our technique falls into the same category as those.
>
> Q: “What reason do you have to hope for this? Why should they be complimentary? Existing optimization techniques are based on careful design and coupling of gradients or surrogate gradients, with specific learning rate schedules. Arbitrarily mixing the two doesn't seem to be theoretically well-motivated.”
> A: Again, neither our starting point nor the goal of this paper is on theory, just like most of the prevalent techniques. Our intuition is supported by the thorough experiments, not by the proof. We also point out that none of the current optimization techniques can be proved to work under the general neural network setting, without unrealistic assumptions.
>
> Q:  “It would be great to provide some intuition for this”.
> A: The intuition is that when the network is deep, the original gradient in the low layers will become very small or very large because of the multiplicative effect of the gradient of the upper layers, which is called gradient vanishing or explosion phenomenon. The layer-wise gradient normalization can counteract this negative effect automatically, maintaining the gradient magnitude per layer as a constant, so that the information can still backprop to the bottom layers.
>
> Q: “This suggests that there is no theoretical motivation to use NGD with an adaptive step size over AdaGrad.”
> A: Yes, you are correct, our motivation is not from theory but from the practical observation. Please also see the response to the previous question.
>
> Q: “How are results that go from quasi-convex to upper semi-continuous quasi-convex an extension?”
> A: It is indeed a typo. We missed “differentiable”. We meant to say “NGD can find a \eps-optimal solution....when the objective function is differentiable quasi-convex.” Kiwiel (Kiwiel, 2001) extended NGD for upper semi-continuous (not necessarily differentiable) quasi-convex objective functions.
>
> Q: “There should be a reference for RMSProp.”
> A: We will cite Geoff Hinton’s lecture note, as there is no formal publication on this method.
>
> Q: “Why is layer parametrization (and later on normalization) a good way idea?”
> A: We repeat the intuition here that when the network is deep, the original gradient in the low layers will become very small or very large because of the multiplicative effect of the gradient of the upper layers. The layer-wise gradient normalization can counteract this negative effect automatically, maintaining the gradient magnitude as a constant, so that the information can still backprop to the bottom layers.
>
> Q: “This seems insufficient for an empirical paper, since most problems often involve some amount of hyperparameter tuning. How sensitive is each method to the choice of hyperparameters? What about the impact of initialization?”
> A: The goal of the experiments is to compare the performance of the existing algorithms and their gradient normalized counterpart. We believe that as long as they are using the same parameter settings, the comparison is fair. Although hyperparameters tuning is orthogonal to the goal of our paper, we actually searched over the learning rate for Adam or other parameters, please see Sec 4.2 of the revision. Besides, we included the mean and variance of the performance over 5 runs for each method, each ResNet and each dataset with random initialization in Sec 4.2.
>
> Q: “NG_{unit} is never motivated.”
> A: Thanks for pointing this out! We have clarified this in the revision and also rename the method. The new method is a variant when the normalization is relaxed to not be strictly 1. We empirically find that it helps improve the generalization performance in Sec 4.2.
>
> Q: “Why are SGD and Adam used for MNIST but not on CIFAR?”
> A: Interestingly, even the Table 1 in the first submission exactly shows the SGD and Adam results on CIFAR10. We also added the result on CIFAR100 in the revision.
>
> Q: “are these parameters chosen from the training set or the test set?”
> A: They are chosen from validation set, which is clarified in the revision.
>
> Q: “Why isn't Adam tested on ImageNet?”
> A: We also included the Adam result on ImageNet in the revision. In fact, as a common wisdom, CNN is the basic model for ImageNet and SGD+momentum is usually better than Adam when using CNNs. That’s why we did not use Adam in the first version. We confirm this result in revision.

---

### Official Review · AnonReviewer3 · 2017-11-27
**A very interesting and important paper**

**Rating:** 9
**Confidence:** 5

**Review:**

This paper illustrates the benefits of using normalized gradients when training deep models.
Beyond exploring the "vanilla" normalized gradient algorithm they also consider adaptive versions, i.e., methods that employ per block (adaptive) learning rates using ideas from AdaGrad and Adam.
Finally, the authors provide a theoretical analysis of NG with adaptive step-size, showing convergence guarantees in the stochastic convex optimization setting.

I find this paper both very interesting and important.
The normalized gradient method was previously shown to overcome some non-convex phenomena which are hurdles to SGD, yet there was still the gap of  combining NG with methods which automatically tune the learning rate.

The current paper addresses this gap by a very simple (yet clever) combination of NG with AdaGrad and Adam, and the authors do a great job by illustrating the benefits of their scheme by testing it over a very wide span of deep learning
models. In light of their experiments it seems like AdamNG and NG should be adopted as the new state-of-the-art methods in deep-learning applications.

Additional comments:
-In the experiments the authors use the same parameters as is used by Adam/AdaGrad, etc..
Did the authors also try to fine tune the parameters of their NG versions? If so what is the benefit that they get by doing so?
-It will be useful if the authors can provide some intuition about why is the learning rate  chosen per block for NG?
Did the authors also try to choose a learning rate per weight vector rather than per block? If so, what is the behaviour that they see.
-I find the theoretical analysis a bit incomplete. The authors should spell out the choice of the learning rate in Thm. 1 and compare to AdaGrad.

---

> ### Author Response · Authors · 2018-01-05
> **Response to AnonReviewer3**
>
> We first thank the reviewer for the valuable feedback. The responses to your questions are as follows:
>
> Q: “In the experiments the authors use the same parameters as is used by Adam/AdaGrad, etc. Did the authors also try to fine tune the parameters of their NG versions? If so what is the benefit that they get by doing so?”
> A: We keep using the same parameters for both the normalized and original version, to make the comparisons fair. Otherwise, if we change the parameters in the normalized version, it is hard to tell whether the effect is due to the normalization or parameter tuning.
>
> Q: “It will be useful if the authors can provide some intuition about why is the learning rate  chosen per block for NG?
> A: “Block” in the neural network scenario means “layer”. So our method is a layer-wise normalization approach. The intuition is that when the network is deep, the original gradient in the low layers will become very small or very large because of the multiplicative effect of the gradient of the upper layers, known as gradient vanishing or explosion phenomenon. The layer-wise gradient normalization, which can also be interpreted as layer-wise learning rate, can counteract this negative effect automatically, maintaining the gradient magnitude as a constant, so that the information (error) can still backprop to the bottom layers.
>
> Q: “Did the authors also try to choose a learning rate per weight vector rather than per block? If so, what is the behaviour that they see.”
> A: If we take all the variables of a neural network as a long vector, normalizing the gradient layer-wisely somehow has already changed the direction of this vector. And if we normalize by each weight vector, making the granularity of the normalization even finer, we are afraid the direction change will be more severe. Consider the extreme case of normalizing each dimension, which is equivalent to choose the sign of each coordinate of the gradient. We believe this would jeopardize the algorithm significantly. However, we feel it makes more sense to address the differences of the gradient magnitude between layers, rather than changing the relative values of weights within the same layer.
>
> Q: “The learning rate in Thm. 1”
> A: This learning rate is chosen to get through the proof under the convex setting. However, we should point out that in our experiments, where the objective function is no longer convex, it is unclear whether this learning rate would still provide convergence guarantee.

---

### Official Review · AnonReviewer4 · 2017-12-14
**Paper is good start for a research question, but is insufficient for a research publication.**

**Rating:** 2
**Confidence:** 5

**Review:**

This paper proposes a variation to the familiar AdaGrad/Adam/etc family of optimization algorithms based a gradient magnitude normalization. More precisely, the components of the gradient are split into blocks (one block per layer), and each block is normalized by its L2 norm. The concatenation of these normalized gradients are used in place of the standard gradient in AdaGrad/Adam/SGD. The authors find the resulting optimizer performs slightly better than its competitors on four tasks.

I feel this paper would be much stronger focusing extensively on one or two small problems and models, providing insight into how normalization affects optimization, rather than chasing state-of-the-art numbers on a variety of datasets and models. I believe the significance and originality of this work to be lacking.

## Pros ##

The paper is easy to follow. The algorithm and experiment setups are clearly explained, and the plots are easy to understand. I appreciate the variety in experimental setups. The authors provide a proof of convergence for the AdaGrad variant on convex functions.

## Cons ##

The paper fails to provide new insights to the reader. It succeeds in asking a question (how do normalized gradients impact training of neural networks?), but fails to add theoretical or empirical knowledge that furthers the field. While effectively changing the geometry of the problem, no motivation (theoretical or intuitive) is given as to why this normalization scheme should be effective.

From the empirical side, the authors compare the proposed optimizers on many datasets and models, but concerningly only using the baselines' default hyperparameters. Even ADAM, a supposedly "hands-free" optimizer, has been shown to vary greatly in performance when its hyperparameters are well chosen (https://arxiv.org/abs/1705.08292). This is simply unfair to the baselines, and conclusions cannot meaningfully be drawn from this alone. In addition, different tasks use different optimizers, which strikes me as odd, and no error bars are added to any plots.

From the theoretical side, the authors show a convergence bound that is minimized when the number of blocks is one. This, however, is not what the authors use in experiments, and no reasoning about the choice of blocks == network layers is given.

## Specific comments ##

p1: "Gradient computation is expensive" is not a good justification. All empirical risk minimization, convex or not, requires a full pass over the dataset. Many convex problems outside of ERM involve very expensive gradient computations.

p1: "These two challenges indicate that for each iteration, stochastic gradient might be the best practical first order information we can get". See loads of work in approximate second-order methods that show otherwise! Hessian-free Optimization, K-FAC, Learning to Learn Gradient Descent, ACKTR's use of Kronecker-factored Trust Region.

p2: You may want to reference Layer-Specific Adaptive Learning Rates for Deep Networks (https://arxiv.org/pdf/1510.04609.pdf), as it appears relevant to the layer-wise nature of your paper.

p2: "Recently, provably correct algorithms..." I'm fairly confident that Adam and RMSProp lack provable correctness. You may want to soften this statement.

p3: The expression being minimized is the sample risk, rather than the expected risk.

p5: The relationship between NG and NG_{UNIT} is confusing. I suggest keeping only the vanilla method analyzed in this paper, or that the second method be better motivated.

---

> ### Author Response · Authors · 2018-01-05
> **Response to AnonReviewer4 (part 1)**
>
> We first thank the reviewer for the valuable feedback. The responses to your questions are as follows:
>
> Q: “I feel this paper would be much stronger focusing extensively on one or two small problems and models, providing insight into how normalization affects optimization, rather than chasing state-of-the-art numbers on a variety of datasets and models.”
> A: Thanks for the suggestion! We indeed put more results (figures and tables) and analysis in Sec 4.2, by investigating the CIFAR10 and 100 datasets. Interestingly, this point of view strikingly contradicts with Reviewer 2’s, who requires “Every state-of-the-art algorithm should be used as a point of comparison on every dataset considered.”
>
> Q: “The paper fails to provide new insights to the reader. It succeeds in asking a question (how do normalized gradients impact training of neural networks?), but fails to add theoretical or empirical knowledge that furthers the field. While effectively changing the geometry of the problem, no motivation (theoretical or intuitive) is given as to why this normalization scheme should be effective.”
> A:  The intuition is that when the network is deep, the original gradient in the low layers will become very small or very large because of the multiplicative effect of the gradient of the upper layers, which is called gradient vanishing or explosion phenomenon. The layer-wise gradient normalization, which can also be interpreted as layer-wise learning rate, can counteract this negative effect automatically, maintaining the gradient magnitude as a constant, so that the information can still backprop to the bottom layers. We agree that our intuition is not strictly supported by theory, but this is also true of many of the effective approaches in deep learning, such as batch normalization, layer normalization, weight normalization and gradient clipping. Lacking theory does not prevent those method becoming prevalent.
>
> Q: “This is simply unfair to the baselines, and conclusions cannot meaningfully be drawn from this alone.”
> A: The goal of the experiments is to compare the performance between the existing algorithms and their gradient normalized counterpart. Hyperparameter tuning is orthogonal to our goal. We believe that as long as they are using the same parameter settings, the comparison is fair.  In fact, in the CIFAR10/100 and ImageNet experiment, we largely adopted the parameter settings in [1], where the model championed the ImageNet 2015 challenge, except for the layer number, mini-batch size and GPU number (we don’t have that many GPUs). This should be considered a very strong baseline. Those parameters were well tuned by other researchers and we don’t see the necessity for re-tuning. That said, we actually searched over the learning rate for Adam or other parameters, please see Sec 4.2 of the revision.
>
> [1] He et.al.. Deep Residual Learning for Image Recognition. CVPR 2016.
>
> Q: “In addition, different tasks use different optimizers.”
> A: In fact, it is commonly observed that for RNNs, the adaptive step-size method like Adam performs better, while for CNNs, SGD+momentum works much better. That’s why we selected the best baseline optimizers for the specific tasks and compare with their normalized gradient counterpart based on this. We have clarified this point in the revision. We have also added the Adam experiment on CNN with ImageNet data, and confirmed this common observation.
>
> Q: “no error bars are added to any plots”
> A: We have added the means and variances in the tables of  Section 4.2.
>
> Q: “the authors show a convergence bound that is minimized when the number of blocks is one.”
> A: According to what Theorem 1 stated, we agree that our convergence bound is minimized when the number of blocks is one. However, this is not the property of the algorithm. Instead, it is just because of our analysis. In the revision, we have derived the convergence bound in a tighter way so that the optimal number of blocks is not necessarily one. In fact, this is easy to derive. Instead of considering a constant M bounding the full gradient \|F’\|, we must consider a block-dependent constant M_i that upper bounds the corresponding block of gradient \|F‘_i\|. By simply replacing all M by M_i in the proof of Theorem 1, we obtain a convergence bound like O( [D^2\sqrt{Bd}/\eta+ \eta(\sum_i M_i^2)\sqrt{d_i}] / sqrt{T} ). Then, consider a situation where some M_k is much larger than other M_i and some d_h is much larger than other d_i  but h is different from k. For instance, we can have M_k=O(M)>>1 and d_h=O(d) but d_i=M_i=O(1) for other i. Our new convergence bound becomes [D^2\sqrt{Bd}/\eta+ \eta(M^2+B+\sqrt{d})] / sqrt{T}. After optimizing eta, we obtain [D(Bd)^{1/4}\sqrt{M^2+B+\sqrt{d}}] / sqrt{T}. Compared this bound for B=1, which is [DM\sqrt{d}] / sqrt{T}, our bound can be lower, for example, when B<M^2. We have add some discussions on when the new convergence bound when $B>1$ is better than when $B=1$ in the revision.

---

> ### Author Response · Authors · 2018-01-05
> **Response to AnonReviewer4 (part 2)**
>
> Q: “Gradient computation is expensive” is not a good justification.
> A: By saying so, we want to emphasize that the full gradient computation for deep neural networks is unrealistic and we thus often use stochastic gradient. We made this point clearer in the revision.
>
> Q: On other optimization methods.
> A: We have softened our expression in the original text to avoid any confusion. Thanks for the pointing this out!
>
> Q: Additional references.
> A: Thanks for bringing this up, we will cite and discuss these two papers. In a nutshell, the updating rule in “Layer-Specific Adaptive Learning Rates for Deep Network” is different from ours in that they are essentially adding a term to the gradient rather than normalizing it, while the paper https://arxiv.org/abs/1705.08292 focuses more on the bad generalization cases of the adaptive step-size methods, which is orthogonal to our focus.
>
> Q: “I'm fairly confident that Adam and RMSProp lack provable correctness. You may want to soften this statement.”
> A: Yes we agree that there are some flaws in the proof of Adam, even for the convex case. That’s why our special case analysis can only be applied to Adagrad. We have softened this statement in the revision.
>
> Q: “The expression being minimized is the sample risk, rather than the expected risk.”
> A: In machine learning, the ultimate goal is to minimize expected risk, although in practice we can only work on the sample risk instead. Nevertheless, we think this is a minor point. In fact, the expression we wrote can present either sample risk or expected, depending on what is the distribution of \xi in that expectation. If we consider the case where the distribution of \xi in our minimization is simply the empirical discrete distribution corresponding to the finite sample, that expectation will just become the average of risk over samples. We prefer to use expectation instead of finite sum expression because the former is more general and our algorithms and theorem can be both applied to minimizing an expectation, no matter the corresponding distribution is continuous (expected risk) or discrete (sample risk).
>
> Q: “The relationship between NG and NG_{UNIT} is confusing.”
> A: We have clarified this in the revision and also rename the methods. The new method is a  variant when the normalization is relaxed to not be strictly 1. We empirically find that it helps improve the generalization performance in Sec 4.2.

---

### Decision · Program_Chairs · 2018-01-29
**ICLR 2018 Conference Acceptance Decision**

**Decision:**

Reject

**Comment:**

The paper proposes to study the impact of normalizing the gradient for each layer before applying existing techniques such as SG + momentum, Adam or AdaGrad. The study is done on a reasonable number of datasets and, after the reviewers' comments, confidence intervals have been added,  although Table 1 puts results in bold but many of these results are not statistically significant.

The paper, however, lacks a proper analysis of the results. Two main things could be improved:
- Normalization does not always have the same effect but the reasons for it are not discussed. This needs not be done theoretically but a more thorough analysis would have been appreciated.
- There is no hyperparameter tuning, which means that the results are heavily dependent on which hyperparameters were chosen. Thus, it is hard to draw any conclusion.

Regarding the seemingly conflicting remarks of the two reviewers, it all depends on what the paper is trying to achieve. If it tries to show that is it state-of-the-art, then comparing to state-of-the-art algorithms on every dataset is crucial. If it tries to study the impact of one specific change, in this case layer normalization, on the optimization, then comparing to the vanilla version is fine. The paper seems to try to address the latter so it is OK if it is not compared to all the state-of-the-art algorithms. However, proper tuning of existing methods is still required.

Ultimately, a better understanding of layer normalization could be useful but the paper is not convincing enough to provide that understanding. There is no need to increase the number of datasets but it should rather focus on designing setups to test and validate hypotheses.